# *In Vivo* Anti-Alzheimer and Antioxidant Properties of Avocado (*Persea americana* Mill.) Honey from Southern Spain

**DOI:** 10.3390/antiox12020404

**Published:** 2023-02-07

**Authors:** Jose M. Romero-Márquez, María D. Navarro-Hortal, Francisco J. Orantes, Adelaida Esteban-Muñoz, Cristina M. Pérez-Oleaga, Maurizio Battino, Cristina Sánchez-González, Lorenzo Rivas-García, Francesca Giampieri, José L. Quiles, Tamara Y. Forbes-Hernández

**Affiliations:** 1Department of Physiology, Institute of Nutrition and Food Technology “José Mataix Verdú”, Biomedical Research Centre, University of Granada, 18016 Armilla, Spain; 2Apinevada Analytical Laboratory of Bee Products, 18420 Lanjarón, Spain; 3Department of Biostatistics, Universidad Europea del Atlántico, Isabel Torres 21, 39011 Santander, Spain; 4Department of Biostatistics, Universidad Internacional Iberoamericana, Arecibo, PR 00613, USA; 5Department of Biostatistics, Universidade Internacional do Cuanza, Cuito 250, Angola; 6Department of Clinical Sciences, Polytechnic University of Marche, 60131 Ancona, Italy; 7International Joint Research Laboratory of Intelligent Agriculture and Agri-Products Processing, Jiangsu University, Zhenjiang 212013, China; 8Sport and Health Research Centre, University of Granada, C/Menéndez Pelayo 32, 18016 Granada, Spain; 9Research Group on Foods, Nutritional Biochemistry and Health, Universidad Europea del Atlántico, Isabel Torres, 21, 39011 Santander, Spain; 10Research and Development Functional Food Centre (CIDAF), Health Science Technological Park, Avenida del Conocimiento 37, 18016 Granada, Spain

**Keywords:** Aβ, tau, AAPH, oxidative stress, ROS, phytochemical, Alzheimer, tauopathies

## Abstract

There is growing evidence that Alzheimer’s disease (AD) can be prevented by reducing risk factors involved in its pathophysiology. Food-derived bioactive molecules can help in the prevention and reduction of the progression of AD. Honey, a good source of antioxidants and bioactive molecules, has been tied to many health benefits, including those from neurological origin. Monofloral avocado honey (AH) has recently been characterized but its biomedical properties are still unknown. The aim of this study is to further its characterization, focusing on the phenolic profile. Moreover, its antioxidant capacity was assayed both *in vitro* and *in vivo*. Finally, a deep analysis on the pathophysiological features of AD such as oxidative stress, amyloid-β aggregation, and protein-tau-induced neurotoxicity were evaluated by using the experimental model *C. elegans*. AH exerted a high antioxidant capacity *in vitro* and *in vivo*. No toxicity was found in *C. elegans* at the dosages used. AH prevented ROS accumulation under AAPH-induced oxidative stress. Additionally, AH exerted a great anti-amyloidogenic capacity, which is relevant from the point of view of AD prevention. AH exacerbated the locomotive impairment in a *C. elegans* model of tauopathy, although the real contribution of AH remains unclear. The mechanisms under the observed effects might be attributed to an upregulation of daf-16 as well as to a strong ROS scavenging activity. These results increase the interest to study the biomedical applications of AH; however, more research is needed to deepen the mechanisms under the observed effects.

## 1. Introduction

Honey is the most known, famous, and appreciated honeybee product, a complex mixture of nutrients and bioactive compounds with multiple biological activities [1]. There are many different types of honey depending on its botanical origin or even on the honeybee species (although *Apis mellifera* is the most common). Monofloral honeys are those coming in total or mainly from a single type of flower or plant. Any monofloral honey must have a particular compositional and organoleptic profile as well as specific physicochemical and microscopic properties according to the Codex Alimentarius [2]. It has been described that the botanical origin affects not only the flavor and taste of honey but also to its chemical composition and finally its biomedical properties [1,3]. Nowadays, avocado (*Persea americana* Mill.) is an important cultivar in tropical and subtropical world areas. In Spain, around 90% of this species comes from the Andalusia region (mainly Granada and Málaga) where it is additionally an important source for nectar production [4]. Recently, avocado honey (AH) has been characterized concerning physicochemical parameters and different compositional aspects [4]. However, there is still a lack of information regarding the biomedical properties of this monofloral honey.

Avocado pulp and avocado byproducts such as seeds or peels have a high phytochemical content, especially antioxidants, with demonstrated potential neuroprotective effects [5]. Therefore, the intake of foods containing bioactive molecules may help the prevention and management of neurodegenerative diseases such as dementias, including Alzheimer’s disease (AD) [6,7]. AD is the most prevalent type of dementia representing the seventh cause of death in the world. It is currently estimated that 55 million people are affected and that almost 10 million new cases are diagnosed each year [8]. The main features of AD include memory, cognitive, behavioral, and motor decline, all of them leading to a high degree of morbidity and mortality [9]. The pathobiology of AD includes neurotransmission defects, protein misfolding, synaptic alterations, and in a more general context, mitochondrial alterations, oxidative stress, and inflammation. Among the most investigated hallmarks of AD, oxidative stress, amyloid β-peptide, and the deposition of protein tau-associated neurofibrillary tangles are included [9]. Recently, it has been reported that AD incidence is declining in some regions of the world at the time that some risk factors such as a low percentage of high education level in the population or cardiovascular diseases are more effectively prevented [10]. This finding, together with the little or almost no therapeutical arsenal already available against AD, reinforces the need for investigating any potential way to prevent or ameliorate the progression of the disease from the clinical and cellular point of view.

The use of *Caenorhabditis elegans* to investigate the potential benefits of foods and food-derived compounds in AD is well-known since *C. elegans* conserve most of the genes (e.g., APP-related genes) in which mutations are correlated with AD. In addition, *C. elegans* is also an interesting model of learning and memory impairments seen during AD. Therefore, these features make *C. elegans* an interesting model to research AD-related processes [11].

In the present study, a monofloral AH from the south of Spain was analyzed to expand its characterization, focusing on the phenolic profile. Moreover, antioxidant capacity of the honey was assayed both *in vitro* and *in vivo*. Finally, a deep analysis on pathophysiological features of AD, such as oxidative stress, amyloid-β aggregation, and protein-tau-induced neurotoxicity, were evaluated to learn about the possible beneficial effects of eating avocado honey in the prevention of AD.

## 2. Materials and Methods

### 2.1. Reagents and Chemicals

Reagents were purchased from Merck (Darmstadt, Germany), Roche (Basel, Switzerland), Sigma-Aldrich (St. Louis, MO, USA), or Thermo Fisher (Waltham, MA, USA) with analytical standards. Distilled deionized water was obtained from a Milli-Q system (Milford, MA, USA).

### 2.2. Honey Sample Preparation

A demi-standardized AH originally from the south of Granada (Spain) subject to Spanish Protected Designation of Origin (PDO) legislation was commercially obtained [12]. Honey was stored in a dark place at room temperature. Intact product was weighted and directly diluted in double distilled deionized water and stored at −80 °C until use. Honey dilution was made fresh weekly until it was running out or the end of experiments. For *C. elegans* experiments, honey was directly added to the nematode growth medium (NGM).

### 2.3. General Honey Composition

AH was extensively studied regarding water, sugar (glucose and fructose), amino acids (proline, phenylalanine, arginine, and histidine), and ashes (calcium, potassium, sodium, phosphorus, and magnesium) content. The procedure for all determinations was previously described by Serra Bonhevi et al. (2019) [4].

### 2.4. Total Content of Flavonoid and Phenolic Compounds in Honey

Total phenolics (TPC), and flavonoids content (TFC) of AH were assessed by colorimetric procedures following the same protocol previously published by Navarro-Hortal et al. (2022) [13]. In this case, results were presented as milligram (mg) of gallic acid equivalent/kilogram (kg) fresh weight (FW) for phenolics determination or mg of catechin equivalent/kg FW for flavonoids measurement.

### 2.5. Physicochemical Properties and Quality Control Parameters of Honey

Physicochemical properties such as color, electrical conductivity, acidity (Free, lactonic and total), and α-Glucosidase activity were assessed in AH. Similarly, some quality control parameters of AH such as *Persea* spp. pollen, perseitol, and hydroxymethylfurfural content were also evaluated [4].

### 2.6. Honey Antioxidant Activity

Antioxidant activity of AH was determined by FRAP, DPPH, and ABTS methods. Procedures for *in vitro* antioxidant assays were described by Rivas-García et al. (2022) in a previous study [14]. Results were presented as μM trolox/kg FW. Every determination was conducted at least three times and absorbances were measured using a microplate reader (Synergy Neo2 Biotek, Winooski, VT, USA).

### 2.7. Identification and Quantification of Individual Phenolic Compounds by Mass Spectrometry in Honey Extracts

Tentative individual identification and quantification of the compounds was done in an AH extract. The AH phenolic compounds extraction was assessed as defined by Afrin et al. [15]. The identification was performed using UPLC-QTOF-MS/MS and the condition operations were previously described [16]. Molecular ions and fragments were used to identify the phytochemical compounds present in the AH by comparison using a MassLynx V4 software (Waters Laboratory Informatics, Mildford, USA). For quantitative analysis of the phenolic compounds, the instruments and condition operations were carried out as described previously by Sanchez-Hernandez et al. [17]. Retention peak times and fragmentation data in samples were used to quantify the compounds by comparing them with standards.

### 2.8. Maintenance and Strains of C. elegans

All strains of *C. elegans* were obtained from the CGC (Minneapolis, MI, USA) and were housed at 20 °C on solid NGM plates fed with *Escherichia coli* OP50 in an incubator (VELP Scientifica FOC 120 E, Usmate, Italy). The strains used were: N2-Wild type, CL4176 (dvIs27) and CL802 (smg-1), BR5706 (bkIs10), LD1 *skn-1*::GFP (ldIs7); TJ356 *daf-16p*::GFP (zIs356); OS3062 *hsf-1*::GFP (nsEx1730); TJ375 *hsp-16.2p*::GFP (gpIs1); CF1553 *sod-3p*::GFP (mu1s84); and CL2166 *gst-4p*::GFP (dvIs19). Only CL4176 and CL802 were housed at 16 °C. For experiments, a bleaching method was used to obtain age-matched embryos according with standards protocols [18]. Briefly, worms were washed and collected with M9 buffer and embryos were isolated using bleaching solution (sodium hypochlorite 4% and NaOH 0.5 N [20/80; v/v]). Then, embryos were washed three times and dispensed into the experimental plates.

### 2.9. Toxicological Characterization of the Avocado Honey

#### 2.9.1. Short-Term Toxicity Test

Short-term toxicity was assessed to analyze the potential lethal toxic effect of different concentrations of AH and select no lethal concentrations for further experiments [19]. For this purpose, N2-embryos were grown in NGM plates for 48 h. Then, animals were moved to plates with rising concentrations of AH (0, 25, 50, 100, 200, and 300 mg/mL) without food. After 24 h, worms were considered as alive or dead when there was no response to physical stimulus. Results are presented as the average survival percentage after 24 h of exposure from three independent experiments with, at least, 50 worms per treatment. For the rest of the tests, a non-toxic submaximal dose of 100 mg/mL was selected.

#### 2.9.2. Embryotoxicity Evaluation

The effect on embryonic development and viability of AH was evaluated through the embryotoxicity test reported by Luo et al. (2019) with several modifications [20]. Briefly, N2-embryos were isolated and placed in NGM plates with *E. coli* OP50 and 100 mg/mL of AH or vehicle. After 24 h, larvae were scored using a microscope (Motic Inc., LTD. Hong Kong, China). Results are presented as the mean of the relation between larvae found and the number of embryos dispensed from three independent experiments with, at least, 50 embryos per treatment.

#### 2.9.3. Pharynx Pump Rate

The pharyngeal pump rate test was assessed to evaluate the influence of AH in worm metabolism and food intake [21]. For this purpose, embryos were placed in plates with 100 mg/mL of AH or vehicle for 96 h. Then, worms were transferred to new NGM plates to count the number of pharynx pumps per minute using a microscope (Motic Inc. LTD. Hong Kong, China). Results are expressed as the average of pharynx pumps/minute from three independent experiments with 10 worms per treatment.

#### 2.9.4. Body Length Measurement

Body length test was done to evaluate the role of AH in worm growing [22]. Briefly, N2-Wild type embryos were placed in plates with 100 mg/mL of AH or vehicle for 96 h. Then, worms were washed three times using M9 buffer and animal length was analyzed by BioSorter^®^ (Union Biometrica, Belgium, Europe) flow cytometer using the time of flight (TOF) tool. Results are presented as the average of TOF from three independent experiments with, at least, 50 animals per treatment.

### 2.10. Cytosolic ROS Measurement under AAPH-Induced Oxidative Stress

Chemical-agent-induced ROS rise test was done in order to reveal the *in vivo* antioxidant capacity of AH [23]. Briefly, N2-Wild type embryos were placed in plates with 100 mg/mL of AH or vehicle for 48 h. Then, worms were exposed to 2.5 mM of 2,2′-azobis-2-amidinopropane dihydrochloride (AAPH) for 15 min to induce oxidative stress. Then, AAPH was removed and 25 μM 2′,7′-Dichlorofluorescein Diacetate (DCFDA) was added and incubated for 2 h. Finally, green fluorescence intensity as well as TOF signal were measured using a BioSorter^®^ flow cytometer (Union Biometrica, Belgium, Europe). Results are the mean of average of green intensity of fluorescence normalized by TOF signal from three independent experiments with, at least, 1000 worms per treatment.

### 2.11. Amyloid-β Assay

Amyloid-β (Aβ)-induced toxicity tolerance test was done to determine the potential effect of AH against Alzheimer amyloid-related toxicity [24]. For this purpose, we used CL4176, a sensitive temperature strain that expresses human amyloid β1–42 peptide in muscle cells which causes a progressive impairment of the movement until worms become paralyzed. CL802 was used as a negative control. Briefly, embryos from CL4176 Aβ (+) or CL802 Aβ (−) were placed in plates with 100 mg/mL of AH or vehicle for 48 h at 16 °C. Next, worms were temperature-up-shifted to 25 °C for 20 h to induce endogenous Aβ production. Then, worms were counted every 2 h for 12 h. Animals were classified as paralyzed when there was no feedback to physical stimulus but still being alive. Results are expressed as the percentage of non-paralyzed worms from three independent experiments with, at least, 100 worms per treatment.

### 2.12. Tau Protein Induced Toxicity

Tau-protein-induced toxicity tolerance test was done to evaluate the potential effect of AH to front the neurotoxicity related to hyperphosphorylated tau protein (hp-tau) aggregation, a featured aspect of Alzheimer’s disease [25]. For this purpose, BR5706 strain was used, which expresses a constitutive pro-aggregative human Tau protein in neurons, reflecting in locomotion alterations. N2-Wild type was used as a negative control. In this assay, embryos from N2 hp-tau (−) or BR5706 hp-tau (+) were placed in plates with 100 mg/mL of AH or vehicle for 72 h. Then, worms were moved to a slide with M9 to stimulate animal locomotion. WormLab Imaging System (MBF Bioscience, Williston, Vermont, EE. UU) was used to document, track, and analyze worm movement. Wavelength, stretching effort, and swimming speed were selected as demonstrative parameters to evaluate mobility alterations. Results are expressed as the average of wavelength, stretching effort, and swimming speed from three independent experiments with, at least, 80 worms per treatment.

### 2.13. Gene Expression Analysis of Antioxidant and Proteostasis System Components

Different worm strains containing transgenic genes coupled to the green protein fluorescent (GFP) reporter were used to observe the mechanisms under the protective role of AH *in vivo* [24]. The transcription factors studied using different strains were SKN-1/NRF2 (LD1), DAF-16/FOXO (TJ356), and heat shock transcription factor (HSF)-1 (OS3062). Among the downstream targets of the studied transcriptional factors, SOD-3 (CF1553), HSP-16.2 (TJ375), and GST-4 (CL2166) were studied. For this purpose, all strains were placed in plates with 100 mg/mL of AH or vehicle for 48 h. Then, worms were moved to a slide and anesthetized with sodium azide (15 µM). A Nikon DS-Ri2 camera was used to photograph the worms under the GFP filter (Tokyo, Japan). Finally, to analyze the obtained images, the software NIS-Elements BR was used, and the background signal was removed from the analysis (Nikon, Tokyo, Japan). Results are the mean of the intensity of fluorescence of the specific strain analyzed from three independent experiments with, at least, 30 animals per treatment. An exception was made for TJ356 results which are presented as the average of a semi-quantitative scale (cytosolic “1”, intermediate “2”, or nuclear “3”) of *daf-16*::GFP location from three independent experiments. The specific indication for each strain is presented in Table 1.

### 2.14. Statistical Analysis

Kolmogorov–Smirnov test (normality) and Levene tests (variance homogeneity) were applied to all variables. Different statistical tests were employed for normally distributed variables (Student *t* test) and for non-normally distributed variables (Mann–Whitney-U and Kruskal–Wallis tests). At least three independent experiments were done, and the results were presented as the average ± standard deviation (SD) for honey characterization and standard error of the mean (SEM) for *C. elegans* experiments. Significance was considered for *p* < 0.05. SPSS 24.0 was used for statistical analysis (IBM, Armonk, NY, USA).

## 3. Results and Discussion

### 3.1. Honey Composition, Physicochemical Properties and Antioxidant Activity

As can be observed in Table 2, the values obtained for honey composition regarding water, sugar, amino acids, and ashes content were like those previously obtained in an AH from the south of Spain and a similar trend was observed with the physicochemical and quality control parameters studied. Concerning the parameters that help to consider the present honey as a monofloral avocado honey, Bonvehi and coworkers indicated that the percentage of pollen of *Persea americana* should be higher than 20% and perseitol higher than 0.10 g/100 g. In the present study, the percentage of pollen was lower than 20%, but perseitol was over the minimum required; thus, it can be considered that the present honey is a monofloral avocado honey [4]. Total phenolics content (TPC), total flavonoids content (TFC), as well as total antioxidant capacity (TAC) of AH are also represented in Table 2. Only two authors have evaluated TPC, TFC, and TAC for AH. AH evaluated in the present work showed higher values of TPC (863 ± 82 mg GAE/kg FW) in comparison with an AH from Ecuador (682 ± 58 mg GAE/kg FW) but lower in comparison with an AH from the south of Spain (1240 ± 100 mg GAE/kg FW) [4,26]. Regarding TFC, our AH showed higher values (130 ± 4 mg CAE/kg FW) in comparison with those obtained (42.5 ± 12.2 mg CAE/kg FW) in the Ecuadorian AH [26].

Concerning TAC, values obtained in the present work were similar to those obtained by García-Tenesaca et al. (2017) for FRAP (4254 ± 492) and DPPH (841 ± 52 μM TE/kg FW) assays in the Ecuadorian AH [26]. Interestingly, ABTS values obtained in the present AH were higher in comparison with the Spanish AH (2600 ± 110 μM TE/kg FW) [4]. These differences might be influenced by numerous factors such as the geographical location and botanical origin of the pollinated avocado flowers as well as honey storage conditions that could affect the phytochemical composition and TAC [27,28]. A representative picture of *Persea* spp. pollen can be observed in Figure 1.

### 3.2. Identification and Quantification of the Avocado Honey Phenolic Compounds

Chromatograms for both positive and negative ion mode were assessed to identify a total of 15 compounds (Figure 2) and the tentative identification of the compounds is shown in Table 3.

According to the phenolics quantification (Table 4), the most predominant molecule present in our AH was *o*-vanillin (204.5 ± 9.3 μg/L) which is a member of hydroxybenzaldehydes. In the same way, other highlighted compounds such as ellagic (141.9 ± 3.7 μg/L) and ferulic acid (39.2 ± 0.2 μg/L), members of catechols and hydroxycinnamic acids, respectively, were also found in high proportions. To the best of our knowledge, this is the first study which identifies and quantifies part of the phenolic compounds present in avocado honey. Therefore, no information is available to compare data from the present investigation with other avocado honey analysis. Notwithstanding, other monofloral honeys with interesting biomedical properties such as manuka honey have been studied and their phenolic profile has been reported. In this context, two of the most abundant compounds present in the AH of the present study (*o*-vanillin and ellagic acid) were also present in high proportions in a manuka honey from New Zealand. Similarly, phenolic acids such as rosmarinic and coumaric acids were present in very low proportions, whereas rutin and kaempferol 3-O-glucoside concentrations were situated under the limit of quantification for both honeys. In contrast, the second most abundant compound present in the mentioned manuka honey, naringenin, was not possible to be quantified in AH in the present research [23].

### 3.3. Toxicological Study of Avocado Honey in C. elegans

*C. elegans* is an interesting model to evaluate the effect of foods and food by-products in several biomedical application areas. Therefore, the toxicological characterization was assessed in *C. elegans* to evaluate any potential underlying deleterious effect of AH *in vivo*. For this purpose, several tests were applied to analyze the nematocidal potential of AH such as short-term toxicity and embryotoxicity tests, as well their influence on food intake and growth. As can be observed in Figure 3A, worms exposed for 24 h to increasing concentrations of AH (0, 25, 50, 100, 200, and 300 mg/mL) did not exert acute lethal toxicity except in the higher concentration which showed a more reduced survival rate. These results are in concordance with those obtained by Sajid et al. (2012), which showed that higher concentrations of natural honeys from different floral sources exerts nematocidal activity [61]. Therefore, a non-lethal submaximal concentration (100 mg/mL) was chosen for further experiments to also probe potential cytoprotective effects. Next, the influence of AH in the embryos’ development and viability was also evaluated. As shown in Figure 3B, AH did not exert embryotoxicity in the assayed dosage, showing that 100 mg/mL was a suitable concentration for subsequent experiments. Additionally, the effect of AH on pharynx pump rate as well as body length was studied. Traditionally, pharynx pump rate is used in bibliography as an overview of correct food intake. In this case, it can be observed that the pump rate increased 1.1%-fold in comparison with unexposed worms (Figure 3C). In fact, the increase of pharyngeal pump rate was also followed by an increase of 1.2%-fold in worms body length (Figure 3C). These data are in accordance with scientific reports, which shows that feeding is one of the most important determinants of growth in *C. elegans*, indicating that worms with lower pump rates become smaller [62]. Taken together, these results indicate that AH, at the dosage used, did not exert nematocidal activity, opening the door for subsequent studies investigating biomedical properties of this honey by using *C. elegans* as an *in vivo* model.

### 3.4. Effect of AH to Fight AAPH-Induced Oxidative Stress

To analyze the potential biomedical applications of the AH, some tests were performed. Firstly, the tolerance against AAPH-induced ROS production was tested. AAPH is a small molecule which is employed in studies to induce lipid peroxidation. Due to the nature of AAPH decomposition, the carbon radicals generated can react with molecular oxygen to generate peroxyl radicals, producing a stable rate of free radicals. Classically, AAPH has been used mostly as a free radical generator *in vitro* [63,64]. However, *in vivo* evidence of non-lethal toxic concentrations of AAPH mimic the results obtained in cell lines concerning free radical production [65]. Therefore, a non-lethal concentration (2.5 mM) of AAPH was selected in this work according to previous *C. elegans* research [13,14,24,25]. As can be observed in Figure 4, the dosage used of AAPH was effective to induce a measurable ROS-production by DCFDA assay. Additionally, a remarkable lower ROS content in AH-treated worms exposed to AAPH was observed. Regarding scientific literature, to the best of our knowledge, no author has explored the effect of avocado honey against oxidative stress *in vitro* or *in vivo*. Several investigations have described the antioxidant activity of other types of honey (gelam, multifloral, or manuka honey) *in vitro* against different stressors such as AAPH [66], lipopolysaccharide [67], and bovine thrombin [68]. In accordance with the *in vivo* studies, some authors showed promising results of these other types of honey against oxidative stress in *C. elegans* [23] and rodents [69,70,71], but some of them did not directly address ROS-related measurements [69,71]. Additionally, the antioxidant properties of honey have recently been supported by a pilot study in humans in which ten healthy individual consuming two spoons/day of Greek honey for one month presented lower ROS levels in blood compared with the control [72].

The potential effects of some pure compounds present in the studied AH to modulate chemical-induced oxidative stress have been previously investigated. In this context, *o*-vanillin (the most abundant compound in AH) reportedly showed antioxidant activity *in vitro* and *in vivo*. Regarding cell lines experiments, *o*-vanillin exerted a strong capacity to modulate chemically induced oxidative stress through AAPH [73], doxorubicin [74], and Fe^2+^ [75]. Similar results were obtained in rodent models supplemented with o-vanillin to fight oxidative stress induced with paclitaxel [76] or maneb [77]. In a similar way, ellagic acid, the second most abundant compound in AH, was found to reduce the pro-oxidative effects produced by diclofenac [78] and hydrogen peroxide [79] *in vitro* as well as acrylamide [80] in rats. Results from the present investigation indicate that avocado honey has a strong capacity to prevent ROS accumulation *in vivo*. This is consistent with the high content of phenolics and flavonoids as well as with the *in vitro* antioxidant activity exhibited in this work. Such *in vitro* and *in vivo* antioxidant capacity, according to the literature, might be attributed, at least partially, to the most abundant phenolic compounds present in AH such as *o*-vanillin and ellagic acid, which have previously shown cytoprotective effects against several chemically induced oxidative stresses.

### 3.5. Effect of AH against Amyloid-β Induced Toxicity

As far as we know, this is the first study to determine the direct cytoprotective effect of an avocado honey, and the second one to test the effect of a whole honey to palliate the toxic events associated with Aβ aggregation *in vivo*. Aβ deposition is one of the two main histopathological lesions occurring in AD. AD is one of the most common neurodegenerative disorders in which an efficient drug has not been discovered. Therefore, to perform a screening of the protective effect of the honey against Aβ-induced toxicity, the modulatory effect of AH was studied in *C. elegans*. CL4176 strain was selected because it expresses the human Aβ1-42 peptide in muscle cells and classically it is used to evaluate the mechanistic pathways of Aβ1−42 toxicity *in vivo* [81]. In fact, this strain has been extensively used to develop a phenotype of progressive paralysis which is directly related to the expression and aggregation of the Aβ1-42 peptide. As can be seen in Figure 5, Aβ-induced paralysis was delayed by the treatment with AH when compared with the untreated CL4176 positive control. CL802 strain is used as a negative control since this is a phenotypically similar strain from which, after the insertion of the Aβ1−42 related gene, is produced the transgenic, thermically induced paralysis CL4176 strain. The results demonstrate how AH avoided paralysis until 28 h post thermal induction, and at the end of the experiment at 34 h post induction, when positive controls were 100% paralyzed and 60% of AH-treated worms were still non-paralyzed. According to the literature, no studies have been performed yet to directly explore the anti Aβ activity of AH neither *in vitro* nor *in vivo*. However, Navarro-Hortal and co-workers demonstrated the anti-amyloidogenic effect of manuka honey by using a similar approach as that followed in the present investigation [23]. On other hand, Rosli et al. showed an improvement of behavior and cognitive function in an AD rat model treated with a multifunctional composition that included honey into its composition. Authors attributed these actions presumably to the reduction of Aβ-toxicity [82].

To learn about the individual molecules present in AH responsible for the observed effects on Aβ-induced toxicity, several studies are available. Thus, *o*-vanillin has shown very low antiaggregatory-Aβ activity in a discontinuous molecular dynamics simulation in combination with the intermediate resolution protein model [83]. A protective effect of ethyl-vanillin, an analog of o-vanillin, was observed against Aβ1-40-induced neurotoxicity in PC12 cells [84]. Similarly, the treatment with vanillic acid, an oxidized form of o-vanillin, improved learning and memory in rats intraventricularly injected with Aβ1-42 [85]. In a similar way, ellagic acid has been shown to reduce Aβ toxicity [86,87] through the reduction of Aβ oligomerization [87] in Aβ-induced neurotoxicity cell lines. These results were also supported in an Aβ-like rodent model treated with ellagic acid which presented a reduction of Aβ deposition in the hippocampus [88], which was reflected with ameliorated spatial learning and lower memory impairment [88,89]. Results from the present study, plus the scientific literature analysis, suggest that the *in vivo* anti-amyloidogenic activity observed in AH could be attributed, at least in part, to the major phenolic compounds present in this honey. Among these, *o*-vanillin and derivatives, as well as ellagic acid, which have demonstrated a remarkable cytoprotective effect against Aβ-induced toxicity in both *in vitro* and *in vivo* studies, should be considered.

### 3.6. Effect of AH against Hyperphosphorylated Tau Protein-Induced Neurotoxicity

To the best of our knowledge, this is the first study to evaluate the potential cytoprotective effect of an avocado honey to face hyperphosphorylated tau neurotoxicity *in vivo*. Tauopathies are characterized by an aberrant hyperphosphorylation of tau protein which assembles into fibrillar polymers, formatting neurofibrillary tangles (NFT) and causing cell death [90]. In fact, some researchers indicate that the increase of NFT deposition is associated with an accelerated progression of the tauopathies as well as the symptoms due its cytotoxicity [91,92]. Therefore, to elucidate a potential cytoprotective effect of the honey against hp-tau-induced toxicity, the modulatory effect of AH was evaluated in *C. elegans*. For this purpose, BR5706 strain, which presents an alteration of the phosphorylation and aggregation of human Tau protein in neurons, was used [93]. This feature has been associated with synaptic malfunction, axonal transport problems, and neurodegeneration which results in locomotion defects quantifiable using a behavioral worm tracking system. As can be seen in Figure 6, a notorious impairment in locomotive behavior was produced in the positive control compared with the negative control in the three studied parameters. Worms treated with AH presented worse parameters than positive controls for the three investigated markers. These bad results for AH could be attributed to the effect on these parameters due to the high content of free sugar present in the composition of the avocado honey [4]. Several studies in preclinical models have shown that a high sugar diet contributes to increased hp-tau gene expression in a tauopathy model [94] or even in non-tauopathy models [95,96,97]. However, a recent work demonstrated that similar doses to that used in the present study of a honey-like sugar mix or manuka honey were able to impair, in the same grade, several locomotive parameters in the strains used as negative controls or hp-tau (+) [23]. Therefore, the obtained results might be attributed to an impairment of mobility caused by the sugar content in AH instead of an exacerbation of hp-tau neurotoxicity. In addition, stretching effort was reduced by the treatment, indicating that a possible cytoprotective effect on tau-related neurotoxicity might be masked by the locomotive impairment produced by the high free sugar content in AH. For further research, an extensive cytoprotective characterization using a honey-like phenolic compound mix must be done in the strains used to evaluate the potential applications to front hp-tau neurotoxicity.

### 3.7. Gene Expression Analysis of Avocado Honey Treated Nematodes

To start the evaluation of the molecular mechanisms under the effects of AH observed in the present study, strains with the specific GFP-marked gene expression were used. In this context, different transcription factors as well as some of their direct downstream targets were evaluated. The transcription factors studied were DAF-16/FOXO, which present a key role in the modulation of Aβ peptide aggregation [98] and oxidative stress responses [99]. In the same way, SKN-1/NRF2 is involved in xenobiotic detoxification and oxidative stress tolerance [100] as well as the modulation of Aβ-induced toxicity [25]. Additionally, HSF-1 was also evaluated due to its role in the reduction of protein misfolding-related proteotoxicity [101]. Among the downstream targets of the studied transcriptional factors, some genes were also analyzed due to involvement in the antioxidant response (SOD-3), xenobiotic elimination (GST-4), as well as proteostasis regulation (HSP-16.2). As can be seen in Figure 7, the transcription factors *skn-1*::GFP and *hsf-1*::GFP were downregulated, whereas *daf-16*::GFP was upregulated by the treatment with AH. As expected, the downstream targets of the transcription factors inhibited were also downregulated with the exception of *gst-4p*::GFP, which was not affected by the treatment. These results might be explained due the high antioxidant activity demonstrated by AH *in vitro* and *in vivo*, particularly, in terms of a free radical scavenger. *in vitro* studies have shown that different natural compounds with ROS scavenger activity reduced the activity or even the expression of different inducible antioxidant enzymes [102]. In accordance with that mentioned above, honey intake has been proved to reduce the gene expression of nrf2, a mammalian ortholog of skn-1, without negatively affecting the total antioxidant status in rats [103]. In the same way, honey treatment was also able to reduce the downstream target of SKN-1/NRF2, such as mitochondrial SOD activity [69] and gene expression [71], without affecting the GSH/GSSG ratio in honey-treated rats. Regarding *daf-16*, the nuclear localization and activation of this specific transcription factor has been linked to stress resistance [104], which in the present study could be directly implicated in the observed protection of AH in the AAPH and Aβ-induced toxicity tests. Finally, the proteostasis network was also evaluated. In this context, the transcription factor *hsf-1*::GFP as well as their downstream *hsp-16.2p*::GFP were both downregulated. Some researchers have linked a downregulation of hsf-1 and its downstream targets with an aberrant hyperphosphorylation of tau protein, which results in increased toxicity [105,106]. These results could explain, at least in part, the negative effects of AH observed in the tau-induced neurotoxicity test, although the real contribution of AH to exacerbated hp-tau toxicity remains unclear. In summary, AH presented a robust antioxidant activity, both *in vitro* and *in vivo*, probably because of a direct ROS scavenger action, which reduces the needs of certain antioxidant system elements. A first approach was made in the presented work regarding the mechanisms involved in the stress response system. Nonetheless, the direct relevance of the downregulated proteostasis network must be studied in the future.

## 4. Conclusions

The AH studied in the present work exerted a high antioxidant capacity *in vitro*. The toxicological evaluation of the AH reported no deleterious effects in *C. elegans* at the dosages used. Concerning the cytoprotective assays, AH presented strong antioxidant activity *in vivo*, preventing ROS accumulation under AAPH-induced oxidative stress. In the same way, AH exerted a great anti-amyloidogenic capacity, leading to a lower Aβ-aggregation, which is relevant from the point of view of AD prevention. In contrast, AH exacerbated the locomotive impairment in a *C. elegans* model of tauopathy, although the real contribution of AH remains unclear. The mechanisms under the observed effects might be attributed to an upregulation of *daf-16* as well as to a strong ROS scavenging activity of AH. Together, these results increase the interest to study the biomedical applications of honey. Notably, the use of the whole honey, instead of extracts, increases the opportunity to generate results which can be directly attributed to the specific food intake. More research is needed to expand on the mechanisms under the observed effects of AH.

## Figures and Tables

**Figure 1 antioxidants-12-00404-f001:**
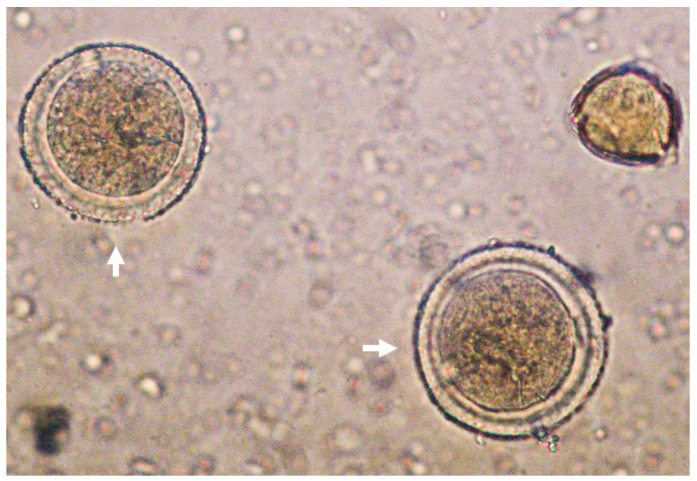
Representative picture of *Persea* spp. pollen (white arrows) under light microscope (40×).

**Figure 2 antioxidants-12-00404-f002:**
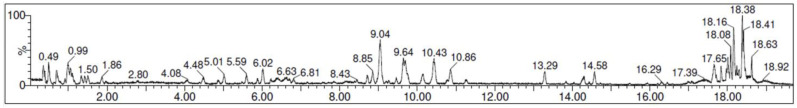
UPLC-QTOF-MS/MS chromatogram of the avocado honey extract.

**Figure 3 antioxidants-12-00404-f003:**
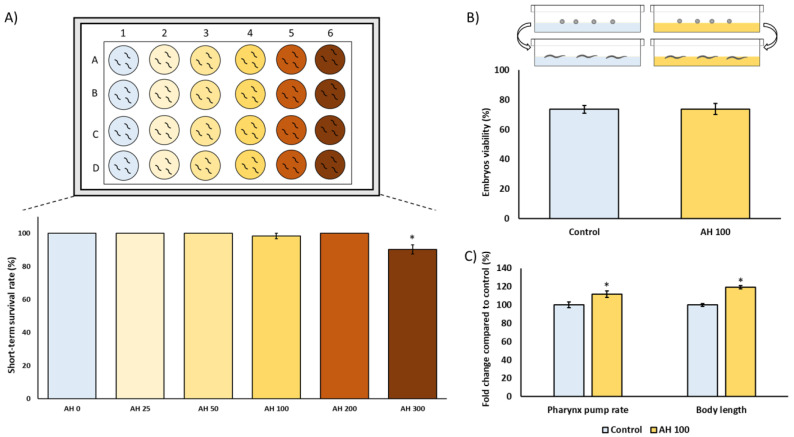
Toxicological characterization of avocado honey (AH) in the *C. elegans* N2-Wild type strain. (**A**) Short-term toxicity test. (**B**) Embryotoxicity evaluation. (**C**) Pharynx pump rate and body length analysis. Results are mean ± SEM. For each parameter, the asterisk (*) represents statistically significant differences with respect to the control (*p* < 0.05). AH concentrations are shown as mg/mL.

**Figure 4 antioxidants-12-00404-f004:**
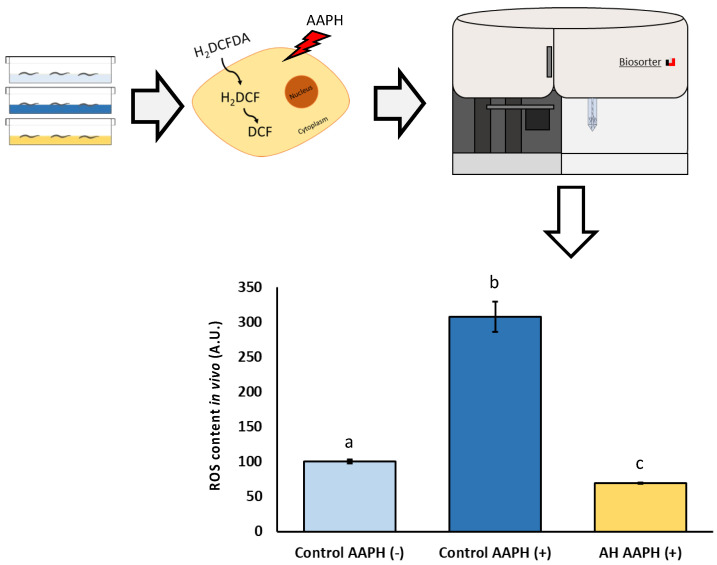
Schematic and analytical representation of the antioxidant analysis of AH against AAPH-induced oxidative stress in *C. elegans*. Results are mean ± SEM. Different lowercase letters means statistically significant differences (*p* < 0.05).

**Figure 5 antioxidants-12-00404-f005:**
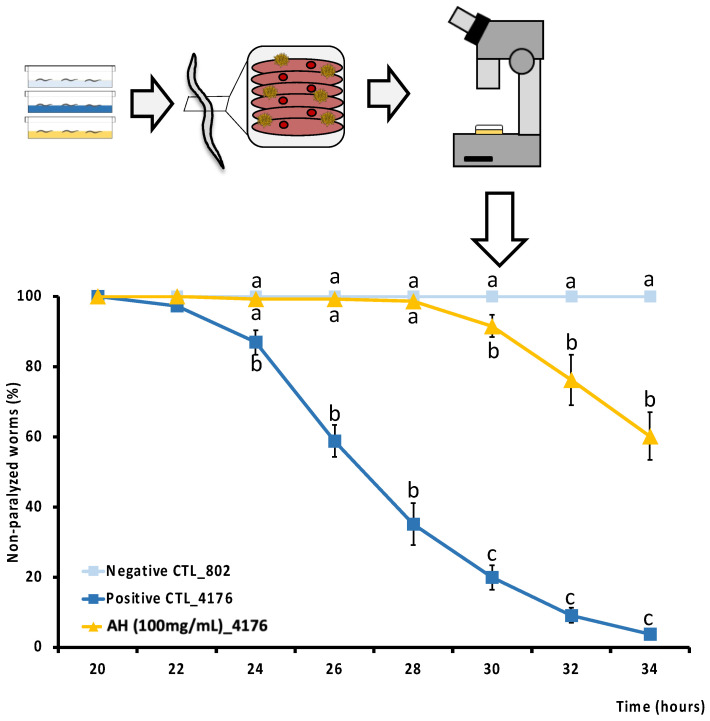
Schematic and analytical representation of the cytoprotective analysis of AH against amyloid-β-induced toxicity in *C. elegans*. Results are mean ± SEM. For each time, different lowercase letters means statistically significant differences between experimental groups (*p* < 0.05).

**Figure 6 antioxidants-12-00404-f006:**
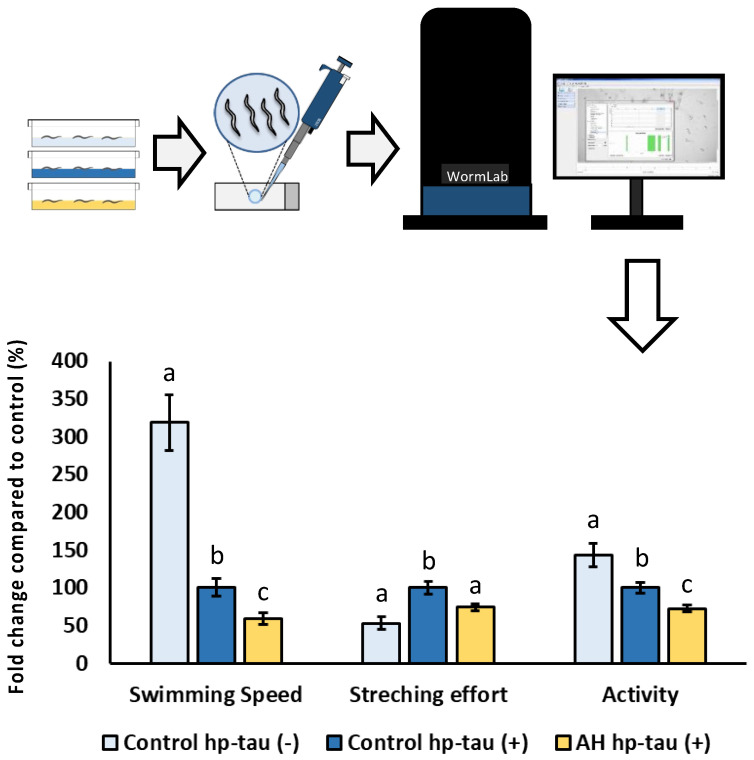
Schematic and analytical representation of the cytoprotective analysis of AH against behavioral changes related to tau protein-induced neurotoxicity in *C. elegans*. Results are mean ± SEM. For each parameter, different lowercase letters mean statistically significant differences (*p* < 0.05).

**Figure 7 antioxidants-12-00404-f007:**
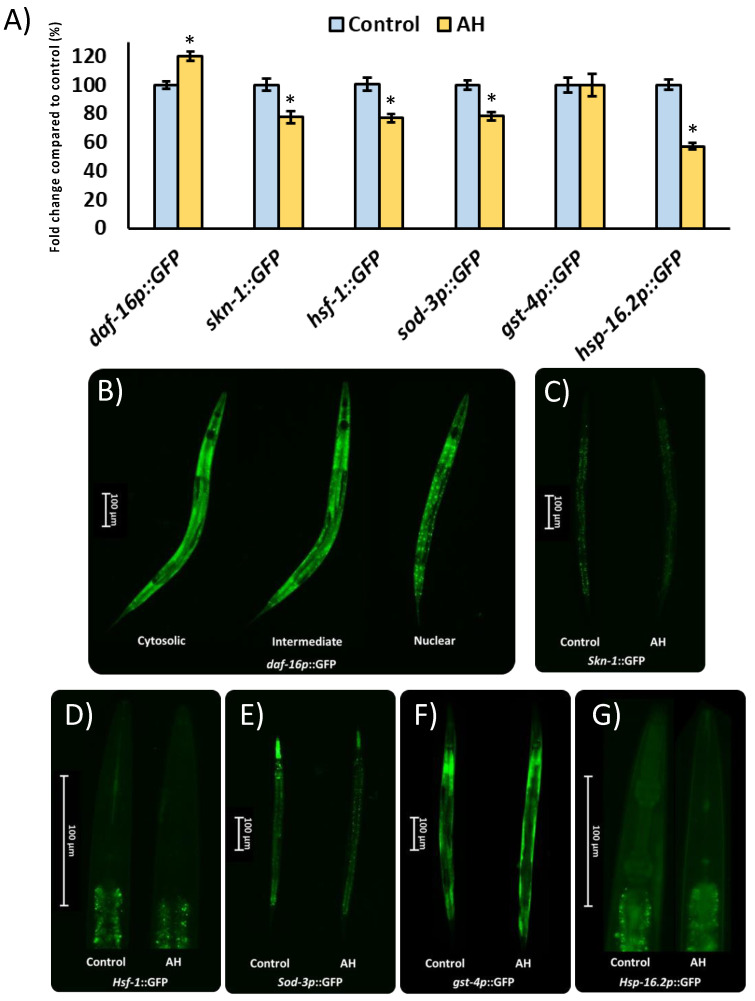
Influence of AH 100mg/mL on gene expression reported by GFP transgenic strains. (**A**) Fluorescence intensity quantification in the specific GFP-reporter strains. (**B**) Representative pictures of the semi-quantitative distribution of *daf-16p*::GPG status. (**C**) Representative pictures for *skn-1*::GFP expression for both groups. (**D**) Representative pictures for *hsf-1*::GFP expression for both groups. (**E**) Representative pictures for *sod-3p*::GFP expression for both groups. (**F**) Representative pictures for *gst-4p*::GFP expression for both groups. (**G**) Representative pictures for *hsp-16.2p*::GFP expression for both groups. Results are mean ± SEM. Asterisk (*) means statistically significant differences (*p* < 0.05) against the control of the same strain.

**Table 1 antioxidants-12-00404-t001:** Compartment location studied in the specific GFP reporter strain.

Strain	Gene-GFP Reporter	Augmentation	ROI
LD1	*skn-1*	10×	Whole worm
OS3062	*hsf-1*	40×	Worm’s bulb
TJ375	*hsp-16.2*	40×	Worm’s bulb
CF1553	*sod-3*	10×	Whole worm
CL2166	*gst-4*	10×	Whole worm

Abbreviations: GFP: Green fluorescent protein; ROI: Region of interest.

**Table 2 antioxidants-12-00404-t002:** Avocado honey composition, physicochemical properties, and antioxidant activity.

Parameters	Mean ± SD	Vmin	Vmax
**General composition**			
Water content (g/kg)	176 ± 8	169	184
Fructose (g/kg)	375 ± 19	358	395
Glucose (g/kg)	306 ± 29	277	335
Fructose/Glucose ratio	1.23 ± 0.06	1.18	1.29
Proline (mg/kg)	410 ± 59	356	473
Phenylalanine (mg/kg)	31.0 ± 9.3	21.2	39.8
Arginine (mg/kg)	28.3 ± 14.1	12.6	39.8
Histidine (mg/kg)	105 ± 65	36	163
Ashes (g/kg)	8.5 ± 3.6	4.5	11.2
Calcium (mg/kg)	71.2 ± 3.9	67.3	75.2
Potassium (mg/kg)	2061 ± 772	1345	2790
Sodium (mg/kg)	84.4 ± 27.2	55.2	109
Phosphorus (mg/kg)	538 ± 89	435	590
Magnesium (mg/kg)	232 ± 84	159	324
**Phytochemical composition**			
Total phenolics content (mg GAE/kg)	863 ± 141	702	969
Total flavonoids content (mg CE/kg)	129 ± 11	113	146
**Physicochemical properties**			
Color (mm scale Pfund)	89.3 ± 7.2	81.0	94.0
Electrical Conductivity (μS/cm)	876 ± 185	750	1090
Free Acidity (meq/kg)	36.0 ± 4.2	32.1	40.4
Lactonic Acidity (meq/kg)	4.2 ± 0.8	3.5	5.1
Total Acidity (meq/kg)	40.8 ± 3.3	37.6	44.2
pH	4.9 ± 0.3	4.5	5.1
**Quality parameters**			
*Persea* spp pollen (%)	11.3 ± 1.5	10.0	13.0
Perseitol (g/kg)	6.9 ± 3.8	3.5	11.0
Hydroxymethylfurfural (mg/kg)	7.8 ± 1.1	6.5	9.3
**Antioxidant activity**			
FRAP (μM trolox/kg)	3740 ± 618	3381	4990
DPPH (μM trolox/kg)	842 ± 186	591	1028
ABTS (μM trolox/kg)	7090 ± 767	6203	7543
**Other**			
α-Glucosidase activity (g sucrose hydrolyzed per 100 g/h)	19.6 ± 3.5	16.3	23.3

Abbreviations: ABTS: 2.2′-azinobis (3-ethylbenzothiazoline-6-sulfonic acid); CE: catechin equivalent; DPPH: 2.2-diphenyl-1-picryl-hydrazyl-hydrate; FRAP: ferric reducing antioxidant power; FW: fresh weight; GAE: gallic acid equivalent; Vmax: maximum value; Vmin: minimum value.

**Table 3 antioxidants-12-00404-t003:** Phenolic compounds identification in the extract of avocado honey.

Tentative Identification	Molecular Formula	Ion Mode	[M-H]	References
**Hydroxycinnamic acids**				
3,4-dicaffeoylquinic acid	C_25_H_24_O_12_	+	517.134	[29,30]
Cinnamic acid	C_9_H_8_O_2_	-	149.059	[29,30,31,32,33]
Caffeic acid	C_9_H_8_O_4_	+/-	181.049/179.034	[29,30,31,34,35,36,37,38,39,40,41,42,43,44,45,46,47,48,49,50,51,52,53,54,55,56,57,58]
Ferulic acid	C_10_H_10_O_4_	+	177.054	[29,30,33,34,35,36,37,38,39,40,41,43,45,46,47,48,49,52,56,57,58,59]
Isoferulic acid	C_10_H_10_O_4_	+	195.065	[49]
*p*-coumaric acid	C_9_H_8_O_3_	+	165.055	[29,30,31,33,34,35,36,37,38,41,43,44,45,46,47,48,49,51,52,53,54,55,56,57,58,59,60]
*o*-coumaric acid	C_9_H_8_O_3_	+	165.055	[33,47]
*m*-coumaric acid	C_9_H_8_O_3_	+	165.055	[33]
**Flavones**				
Quercetin 3-O-rhamnoside	C_21_H_20_O_11_	-	447.093	[34,36]
Apigenin 7-O-glucoside	C_21_H_20_O_10_	+	271.061	[34]
Kaempferol 3-O-glucoside	C_21_H_19_O_11_	-	447.092	[35]
**Hydroxycoumarins**				
Esculin	C_15_H_16_O_9_	-	341.086	[29,34,36]
Scopoletin	C_10_H_8_O_4_	-	191.034	[37,59]
Umbelliferone	C_9_H_6_O_3_	+	163.039	[37]
**Hydroxyphenylacetic acids**				
Phenylacetic acid	C_8_H_8_O_2_	+	137.059	[47]

**Table 4 antioxidants-12-00404-t004:** Phytochemical content in the avocado honey extract.

Phenolic Compounds	Mean (µg/L) ± SD
**Hydroxybenzaldehydes**	
*o*-Vanillin	204 ± 9
**Phenolic acids**	
Ellagic acid	142 ± 4
Ferulic acid	39.2 ± 0.2
m-coumaric acid	18.0 ± 1.3
Rosmarinic acid	2.5 ± 0.3
**Isoflavones**	
Formononetin	9.7 ± 0.9
Glycitein	1.8 ± 0.2
**Flavonol glycosides**	
Rutin	<LOQ
Quercetin	<LOQ
Quercetin-3-O-glucopyranoside	<LOQ
Kaempferol-3-O-glucoside	<LOQ
**Flavanones**	
Naringenin	<LOQ
**Flavanols**	
Epicatechin	<LOQ
**Anthocyanins**	
Chrysanthemin	<LOQ

Abbreviations: LOQ: limit of quantification; SD: standard deviation.

## Data Availability

The data presented in this study are available on request from the corresponding author. The data are not publicly available yet because funded grants are still ongoing.

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
