# Peer review of "In Vivo Anti-Alzheimer and Antioxidant Properties of Avocado (Persea americana Mill.) Honey from Southern Spain"

_antioxidants, 2023, doi:10.3390/antiox12020404_

Round 1

Reviewer 1 Report

An interesting and well-organized study about AH effects in AD, including various experimental procedures.

I have the following comments:

- In 2.8 Section, where the strains of C. elegans are presented, I believe that the authors should add a few words about the unique characteristic of each strain. It will help the readers.

-Moreover, I would like to see in more detail about the reasons for the selection of the C.elegans model in these AD experiments, perhaps in the Introduction.

-Please clarify in the text the numbers of worms used in each experimental group.

-Have the authors used concentrations > 300 mg/mL? Is the survival significantly reduced?

Author Response

REVIEWER 1.

An interesting and well-organized study about AH effects in AD, including various experimental procedures.

AUTHORS: Authors greatly appreciate the very positive comments on the manuscript made by the Reviewer. We appreciate the time and effort that dedicated by the Reviewer to provide such valuable feedback on the manuscript.

I have the following comments:

- In 2.8 Section, where the strains of C. elegans are presented, I believe that the authors should add a few words about the unique characteristic of each strain. It will help the readers.

AUTHORS: Following Reviewer recommendation, a short description of strain features has been added in section 2.13, lines 240 to 244: “The transcription factors studied using different strains were SKN-1/NRF2 (LD1), DAF-16/FOXO (TJ356) and heat shock transcription factor (HSF)-1 (OS3062). Among the downstream targets of the studied transcriptional factors, SOD-3 (CF1553), HSP-16.2 (TJ375), and GST-4 (CL2166) were studied.”

-Moreover, I would like to see in more detail about the reasons for the selection of the C.elegans model in these AD experiments, perhaps in the Introduction.

AUTHORS: According with Reviewer suggestion, a paragraph indicating the interesting use of C. elegans as AD model was added. Lines 83 to 87. “The use of Caenorhabditis elegans to investigate the potential benefits of foods and food-derived compounds in AD is well known since C. elegans conserve most of the genes (e.g. APP-related genes) in which mutations are correlated with AD. In addition, C. elegans is also an interesting model of learning and memory impairments seen during AD. Therefore, these features make C. elegans an interesting model to research AD-related processes.”

-Please clarify in the text the numbers of worms used in each experimental group.

AUTHORS: Done.

-Have the authors used concentrations > 300 mg/mL? Is the survival significantly reduced?

AUTHORS: Since 300 mg/ml was tested and reduced worms’ survival, higher doses were not evaluated.

Reviewer 2 Report

In the manuscript entitled "In vivo anti-Alzheimer and antioxidant properties of avocado (Persea americana Mill.) honey from Southern Spain", Marquez et al. demonstrated the anti-Alzheimer and antioxidant properties of avocado honey. The authors atributed the effects mainly to the chemical compounds within honey. The manuscript is well written and brings new insights on the biological properties of avocado honey. The statistics is well written and the conclusion supports the results. The article could be accepted for publication. 

Author Response

REVIEWER 2.

In the manuscript entitled "In vivo anti-Alzheimer and antioxidant properties of avocado (Persea americana Mill.) honey from Southern Spain", Marquez et al. demonstrated the anti-Alzheimer and antioxidant properties of avocado honey. The authors atributed the effects mainly to the chemical compounds within honey. The manuscript is well written and brings new insights on the biological properties of avocado honey. The statistics is well written and the conclusion supports the results. The article could be accepted for publication.

AUTHORS: Authors greatly appreciate the very positive comments on the manuscript made by the Reviewer. We appreciate the time and effort that the Reviewer dedicated to providing such valuable feedback on the manuscript.

Reviewer 3 Report

Though the research has been particularly performed with avocado honey (AH), authors should include some background, in the Introduction section, about avocado (as a whole: pulp, seeds, peels,…) as a major dietary source of antioxidants and its preventive role in neurodegenerative diseases (as AD).

In the last paragraph of the Introduction section, it is stated that “The use of Caenorhabditis elegans to investigate the potential benefits of foods and food-derived compounds in AD is well known.” If so, bibliographical citation(s) must be provided about it to support the statement.

A well-known antioxidant compound/product, as a positive control, have not used in this research. Authors should explain the reason.

Authors should provide the particular reference of the Avocado honey as a commercial product of Apinevada. Any reader of this potential article in Antioxidants could reproduce this research, using exactly the same product. In addition, authors should provide the date of acquisition of the product, how the product is kept in the original packaging once it has been opened, date of analysis since the initial opening of the commercial packaging, how long aqueous aliquots of the product are stored at -80ºC.

Maintenance and synchronization of C. elegans should be explained in more detail.

It should be considered that honey, especially if it is very pure, may contain traces of pollen inside, which is why it is not recommended for persons allergic to pollen, regardless of the season and the pollen to which they are allergic. Highly manufactured honeys (not so pure) are generally well tolerated. Besides sugar, could the pollen have affected the worm?

Are the negative effects of AH observed in the tau-induced neurotoxicity test offset by its robust antioxidant activity? If the problem is the sugar, and sugar is a main ingredient of honey, is it worth recommending a honey diet as a preventative for neurodegenerative diseases?

Author Response

REVIEWER 3.

AUTHORS: The authors thank the Reviewer for the exhaustive work done on the manuscript, which has undoubtedly improved its quality.

Though the research has been particularly performed with avocado honey (AH), authors should include some background, in the Introduction section, about avocado (as a whole: pulp, seeds, peels…) as a major dietary source of antioxidants and its preventive role in neurodegenerative diseases (as AD).

AUTHORS: according with reviewer request, a brief paragraph has been added.  Line 65 to 67. “Avocado pulp and avocado byproducts like seeds or peels have a high phytochemical content, especially antioxidants, with demonstrated potential neuroprotective effect [5].”

In the last paragraph of the Introduction section, it is stated that “The use of Caenorhabditis elegans to investigate the potential benefits of foods and food-derived compounds in AD is well known.” If so, bibliographical citation(s) must be provided about it to support the statement.

AUTHORS: Added. Line 87

A well-known antioxidant compound/product, as a positive control, have not used in this research. Authors should explain the reason.

Authors: the authors greatly appreciate the Reviewer's comment. In the design of the experiments, authors tried to demonstrate the specific protective role of AH to counteract the deleterious effect of AD hallmarks. In the case of AD-like strains, the negative controls were performed using non pathological strains (CL802 and Wild-type N2), indicating a null disease phenotype in their respective test, being a reliable control group. In the case of oxidative stress tolerance test, authors focused into generate a robust positive control such as AAPH exposition, which caused a boost ROS content in vivo. Therefore, antioxidant compound/product has not been used in this research.

Authors should provide the particular reference of the Avocado honey as a commercial product of Apinevada. Any reader of this potential article in Antioxidants could reproduce this research, using exactly the same product. In addition, authors should provide the date of acquisition of the product, how the product is kept in the original packaging once it has been opened, date of analysis since the initial opening of the commercial packaging, how long aqueous aliquots of the product are stored at -80ºC.

AUTHORS: according with reviewer request, a brief paragraph has been added in order to clarify the commercial origin of the honey and the storage instructions.  Line 102 to 107. “A demi-standardized AH originally from the south of Granada (Spain) subject to Spanish Protected Designation of Origin (DOP) legislation, was commercially obtained. Honey was stored in a dark place at room temperature. Intact product was weighted and directly diluted in double distilled deionized water and stored -80 º until use. Honey dilution was freshly made weekly until that was running out or the end of experiments. For C. elegans experiments, honey was directly added to the nematode growth medium (NGM).”

Maintenance and synchronization of C. elegans should be explained in more detail.

AUTHORS: according with reviewer request, a brief paragraph has been added.  Line 151 to 152 “All strains of C. elegans were obtained from the CGC (Minneapolis, MI, USA) and were housed at 20 °C on solid NGM plates feeded with Escherichia coli OP50 in an incubator (VELP Scientifica FOC 120 E, Usmate, Italy).” Line 157 to 161.  “For experiments, a bleaching method was used to obtain age-matched embryos according with standards protocols. Briefly, worms were washed and collected with M9 buffer and embryos were isolated using bleaching solution (sodium hypochlorite 4% and NaOH 0.5 N [20/80; v/v]). Then, embryos were three times washed and dispensed into the experimental plates.”   

It should be considered that honey, especially if it is very pure, may contain traces of pollen inside, which is why it is not recommended for persons allergic to pollen, regardless of the season and the pollen to which they are allergic. Highly manufactured honeys (not so pure) are generally well tolerated. Besides sugar, could the pollen have affected the worm?

AUTHORS: the authors greatly appreciate the Reviewer's comment. Despite the limited research available in the proposed field, there is classic research which determinate that the ingestion of honey in patients with pollen allergies did not cause obvious reaction after honey intake. (Kiistala et al., 1995. Allergy. doi: 10.1111/j.1398-9995.1995.tb05061.x.). In fact, more recent research demonstrated that the pollen proteins contained in the honey cause allergic reactions to honey only in honey-allergic patient (Bauer et al., 1996. J Allergy Clin Immunol. doi: 10.1016/s0091-6749(96)70284-1.). Therefore, It is unlikely, although not ruled out, that the pollen could negatively affect the worms.

Are the negative effects of AH observed in the tau-induced neurotoxicity test offset by its robust antioxidant activity? If the problem is the sugar, and sugar is a main ingredient of honey, is it worth recommending a honey diet as a preventative for neurodegenerative diseases?

AUTHORS: The results obtained in this research might indicate the possible cytoprotective effect of AH on tau related neurotoxicity might be masked by the locomotive impairment produced by the high free sugar content. Some researchers have described that some C. elegans mutant strains are more sensitive to environmental changes such as an increased osmosis of the NGM (Therrien et al., 2013. PLoS One. doi: 10.1371/journal.pone.0083450.; Vieira et al., 2018. Cell Mol Life Sci. doi: 10.1007/s00018-017-2719-2.). Therefore, AH enrichment of the NGM could increase the total osmotic pressure of the agar plate which, plus an increased sensitivity of the strain, could present a toxic effect. This work opens the gate to increase the visibility of honeybee-products, being preliminary research. Therefore, more research is necessary to confirm these results in animal models such as rodents prior to resolve the proposed question. 

Round 2

Reviewer 3 Report

The manuscript is acceptable for publication.